# Quantum Interference Effects on Information Phase Space and Entropy Squeezing

**DOI:** 10.3390/e21020147

**Published:** 2019-02-05

**Authors:** Abdel-Baset A. Mohamed, Shoukry S. Hassan, Rania A. Alharbey

**Affiliations:** 1College of Science, Prince Sattam Bin Abdulaziz University, Al-Aflaj 11942, Saudi Arabia; 2Faculty of Science, Assiut University, Assiut 71515, Egypt; 3Mathematics Department, College of Science, University of Bahrain, P.O. Box 32038, Zallaq 1051, Bahrain; 4Mathematics Department, Faculty of Science, King Abdulaziz University, P.O. Box 42696, Jeddah 21551, Saudi Arabia

**Keywords:** Wehrl entropy, entropy squeezing, quantum interference, 2-photon transition, squeezed vacuum reservoir

## Abstract

Wehrl entropy and its density are used to investigate the dynamics of loss of coherence and information in a phase space for an atomic model of two-photon two-level atom coupled to different radiation reservoirs (namely, normal vacuum (NV), thermal field (TF) and squeezed vacuum (SV) reservoirs). Particularly, quantum interference (QI) effect, due to the 2-photon transition decay channels, has a paramount role in: (i) the atomic inversion decay in the NV case, which behaves as quantum Zeno and anti-Zeno decay effect; (ii) the coherence and information loss in the phase space; and (iii) identifying temporal information entropy squeezing. Results are also sensitive to the initial atomic state.

## 1. Introduction

Spontaneously generated coherence or quantum interference (QI) process occurs (among other systems) in atomic systems with multi-level structure. it is a result of coupling between two, or more, indirect photon emission channels [1,2]. Spectroscopically, the QI process has shown its role in many experimental quantum optical phenomena. These include, among others, resonance fluorescence linewidth narrowing (i.e., reduction of radiative decay rate) [3] and zero absorption spectrum of a weak field probing atomic media (i.e., transparent atomic medium) [4,5,6,7]. This has its implications, within quantum information processing, to transform information via a probe pulse without any distortion, as well potential applications in other phenomena in non-linear optics [8], e.g., lasing without inversion, slowing and storing light [9,10], and superluminal light propagation ([11] and references therein).

For the particular system of a dissipative two-photon two-level atomic structure in the presence of a broadband squeezed vacuum (SV) radiation reservoir of different configurations, QI takes place via the two indirect photon transition channels through a set of dense intermediate atomic states [12,13,14].

In the present article, we study the dynamics of the 2-photon 2-level atom model [14] regarding effects of the radiation reservoir and QI parameters on the appearance of quantum correlations, loss of coherence and information in the phase space. Particularly, we investigate the non-negative quasi-probability distribution functions [15,16], Husimi Q-function [15], Wehrl entropy [17,18] and Boltzmann entropy squeezing [19,20,21].

The paper is organized as follows. In Section 2, we present the model Bolch equations for the mean atomic variable, together with the mathematical formulas for the Wehrl density and entropy, and Boltzmann entropy squeezing. In Section 3, we present the computational results for the atomic dynamics, as well for the Wehrl entropy and entropy squeezing, followed by a summary of the results in Section 4.

## 2. Model Equations and Mathematical Formulas

### 2.1. Model Bloch Equations

The energy level scheme of the 2-photon 2-level atom with upper |↑〉 and lower |↓〉 states of the same parity with the energies $\hbar \omega_{1}$ and $\hbar \omega_{2}$, respectively, is shown in Figure 1. The intermediate states {|l〉} of energies $\left\{\hbar \omega_{l}\right\}$ are directly coupled to the state |↑〉 or |↓〉, or both through direct-dipole transition. There is no direct-dipole transition between the states |↑〉 and |↓〉 and hence no direct one-photon transition process [12]. We consider the cascade configuration where $\omega_{1}>\omega_{l}>\omega_{2}$ for all *l* (as shown in Figure 1).

In the general case, of the presented model coupled to SV radiation reservoir, the model Bloch equations for the averaged net atomic operators $r_{i}=\langle {\hat{r}}_{i}\rangle$, i=z,+,- (after tracing over the SV radiation reservoir state and adiabatically eliminate the intermediate states {|l〉}—see [12] for rigorous derivation) are put in the normalized matrix form as:(1)$$\begin{array}{ccc} \dot{R}=AR+B,\;A & = & \left(\begin{array}{ccc} a_{1} & f & f\\ -f & a_{2} & 0\\ -f & 0 & a_{2}^{*} \end{array}\right),\;B=\left(\begin{array}{c} -1\\ b_{1}\\ b_{1}^{*} \end{array}\right) \end{array}$$ where $R=[r_{z},\;r_{-},\;r_{+}]$, $a_{1}=-\left[N_{1}+M\cos2\omega_{p}^{{}^{\prime }}\tau \right]$, $a_{2}=-\left[N_{1}+i\omega_{12}-2Me^{2i\omega_{p}^{{}^{\prime }}\tau }\right]$ and $b_{1}=-f\left(N_{1}+2Me^{-2i\omega_{p}^{{}^{\prime }}\tau }\right)$.

The operators ${\hat{r}}_{i}$ are the spin-1/2 Pauli operators,
(2)$$\begin{array}{ccc} r_{z} & = & \left|↑\rangle \langle ↑\right|-|↓\rangle \langle ↓|,\;r_{-}=|↓\rangle \langle ↑|,\;r_{+}=|↑\rangle \langle ↓|. \end{array}$$

The rest of the symbols in Equation (1) are: $\dot{R}=\frac{d}{d\tau }R$, with normalized time τ=γt: $\gamma =\sum_{l}\left(4/3\hbar c^{3}\right)|p_{1l}{|}^{2}\omega_{1l}^{3}=\sum_{l}\left(4/3\hbar c^{3}\right){\left|p_{l_{2}}\right|}^{2}\omega_{l_{2}}^{3}$ is the total decay rate from |1〉→{|l〉} or from {|l〉}→|2〉 (taken equal [12,13]), $\omega_{12}=\left(\omega_{1}-\omega_{2}\right)/\gamma$ is the normalized atomic transition frequency, $\omega_{p}^{{}^{\prime }}=\omega_{p}/\gamma$ is the normalized carrier frequency of the SV radiation reservoir field, $f=\gamma^{12}/\gamma$ is the QI normalized parameter, where $\gamma^{12}=\sum_{l}\left(4/3\hbar c^{3}\right)\left(p_{1_{1}}.p_{l_{2}}\right)\omega_{lk}^{3}$ is the total QI effect between the channels |k〉↔|l〉; k=1,2 with $p_{1_{l}},p_{l_{2}}$ the relevant dipole matrix elements between the states (|1〉,|l〉) or (|2〉,|l〉).

Maximum QI effect occurs with parallel or anti-parallel dipole matrix elements of $p_{1_{l}}$ and $p_{l_{2}}$, so f=+1 or -1, respectively. The parameter $N_{1}=1+2N$, where *N* is the average photon number of the SV field, and $M=\sqrt{N(N+1)}$ is the degree of squeezing (taken real and of maximum value). The special cases of (M=N=0) and (N≠0,M=0) represent the normal vacuum (NV) and thermal field (TF) radiation reservoirs, respectively.

The main physical assumptions involved in the derivation of the model Equation (1) are (see [12]):

(i) The intermediate dense states {|l〉} have no initial population and are of the same parity, hence no transition or interference effect occurs between these states.

(ii) The SV parameters *N*, *M* are assumed the same at all transition frequencies: $N\left(\omega_{al}\right)=N\left(\omega_{bl}\right)=N$; $M\left(\omega_{al}\right)=M\left(\omega_{bl}\right)=M$, for all *l*.

(iii) The usual Markovian approximation and the “2-photon” rotating wave approximation are adopted (i.e., terms in $e^{\pm 4i\omega_{p}^{{}^{\prime }}\tau }$ are discarded compared with terms in $e^{\pm 2i\omega_{p}^{{}^{\prime }}\tau }$).

### 2.2. Mathematical Formulas

#### 2.2.1. Wehrl Density and Entropy

Wehrl density [17] is a powerful tool giving insight into evolution and in the search of quantum correlations of quantum systems. It is a measure for the extraction of phase information, in addition to the analysis of the purity and the entanglement degree of the system. Among other used of Wehrl entropy, it is used as a discriminator between different kinds of superpositions and of statistical mixtures (see [22,23], and references therein). The Wehrl entropy of a quantum state $\hat{\rho }$ is defined as [17]
(3)$$\begin{array}{ccc} S_{\rho } & = & -\int_{\Omega }H_{\rho }\left(\mu \right)\;\ln H_{\rho }\left(\mu \right)\;d\mu , \end{array}$$ where *H* is the Husimi quasi-distribution function $H_{\rho }\left(\mu \right)=\langle \mu |\hat{\rho }|\mu \rangle$ [15], with $\int_{\Omega }H_{\rho }\left(\mu \right)d\mu =1$, and Ω represents the total phase space. Mathematically speaking, it is positive (as the von Newman entropy) and of unit minimum value [24]. For physical problems including spin, the spin coherent state representation is the adapted description due to the algebraic characteristics of the angular momentum operator *J*. The coherent state |μ〉, for a defined angular momentum *j*, is identified by the angles (θ,ϕ) [25] as
(4)$$|\theta ,\phi \rangle =\sum_{k=-j}^{k=j}\frac{e^{i(j-k)\phi }}{2^{j}}\sin^{j}\theta \coth^{k}\frac{\theta }{2}\sqrt{\left(\;\;\begin{array}{c} 2j\\ j-k \end{array}\;\;\right)}|j,k\rangle .$$

In the case of the 2-level system ($j=\frac{1}{2}$), this state is the Bloch coherent state with dμ=sinθdθdϕ.

The Husimi-function is defined as [15]:(5)$$\begin{array}{c} \;\;\;\;\;\;\;\;\;\;\;\;\;\;\;\;\;\;\;\;H\left(\theta ,\phi ,t\right)=\frac{1}{2\pi }\langle \theta ,\phi |{\hat{\rho }}_{A}\left(t\right)|\theta ,\phi \rangle =\frac{1+\chi }{4\pi }, \end{array}$$ where $\chi =r_{z}\cos\theta +\left(r_{x}\cos\phi +r_{y}\sin\phi \right)\sin\theta$, $r_{x}=\frac{1}{2}\left(r_{+}+r_{-}\right)$; $r_{y}=\frac{1}{2i}\left(r_{+}-r_{-}\right)$ and ${\hat{\rho }}^{A}\left(t\right)$ is the reduced density matrix of the atom. Therefore, the Wehrl entropy is given by:(6)$$\begin{array}{ccc} S(t) & = & \int_{0}^{2\pi }\int_{0}^{\pi }D\left(\theta ,\phi ,t\right)\;\sin\theta \;d\theta d\phi , \end{array}$$ where D(θ,ϕ,t) represents the Wehrl density (WD)
(7)$$\begin{array}{c} D(\theta ,\phi ,t)=-H(\theta ,\phi ,t)\;\ln[H(\theta ,\phi ,t)]. \end{array}$$

Thus, Equation (6) takes the form,
(8)$$\begin{array}{ccc} S(t) & = & \ln\left(4\pi \right)-\sum_{n=1}^{\infty }\int_{0}^{2\pi }\;\;\int_{0}^{\pi }\frac{\chi^{n}\left(\chi +1\right)}{\left(4\pi n!\right)}\;\sin\theta \;d\theta d\phi . \end{array}$$

The Wehrl entropy and the Wehrl density, are used to explore the coherence loss of a qubit state and the information loss of its density matrix in a phase space.

#### 2.2.2. Entropy Squeezing

The probability distributions, for any quantum state, for two possible outcomes of measurements of the spin-1/2 Pauli operators ${\hat{r}}_{i}(i=x,y,z$) are calculated by: $\beta_{j}\left({\hat{r}}_{i}\right)=\frac{1}{2}\left[1+{\left(-1\right)}^{j}r_{i}\right]\left(j=1,2\right)$, where $r_{i}$ is the expectation value of the operator ${\hat{r}}_{i}$. The corresponding information entropy of the Pauli operators $r_{i}$ for a 2-level atom system is given by [19,21]:(9)$$\begin{array}{ccc} I(r_{i}) & = & \sum_{j=1}^{2}\beta_{j}\left(r_{i}\right)\ln\left[\beta_{j}\left(r_{i}\right)\right],\;\mathrm{with},\;\sum_{i}\delta I\left(r_{i}\right)\geq 4. \end{array}$$ where $\delta I\left(r_{i}\right)\equiv \exp\left[I\left(r_{i}\right)\right]$. The fluctuations in the atomic components $r_{i}$ are squeezed if the information entropy $I(r_{i})$ satisfy the condition:(10)$$\begin{array}{c} E_{i}=\delta I\left(r_{i}\right)-\frac{2}{\sqrt{\delta I(r_{z})}}<0.\;i=x,y,z. \end{array}$$

## 3. Computational Results

### 3.1. Atomic Behavior

Two remarks are noted about Equation (1): (i) the time-dependent harmonic coefficients are due to the presence of the SV reservoir (M≠0); and (ii) the coupling of the atomic variables, $r_{z,\pm }\left(t\right)$, is wholly due to the QI process (f≠0).

Exact analytical solutions of the non-autonomous Bloch equation (Equation (1)) are derived in [14] for the two special cases (M=0,f≠0) and (M≠0,f=0), namely for the thermal field (TF) reservoir with QI effect and the SV reservoir in the absence of QI, respectively. In the general case (M,f≠0), perturbative solutions of Equation (1) are derived in [14] with perturbation order parameter $\epsilon ≃M/\omega_{p}^{{}^{\prime }}\ll 1$.

Here, we resort to numerical solution of the model Bloch equation (Equation (1)) to examine the effects of the QI parameter, *f*, on the evolution of the atomic variables in this section. To observe the QI effect, we choose the atomic transition frequency normalized to (γ); $\omega_{12}∼10$, which is suitable for Rydberg atomic systems(of very large dipole moments and small radiative decay constants). Further, we take $\omega_{p}^{{}^{\prime }}≃\omega_{12}$ (we found no noticeable effect in the non-resonant case $\omega_{p}^{{}^{\prime }}\neq \omega_{12}$).

**(i)** 
**The NV case (N=M=0)**
In this case, Bloch equation (Equation (1)) reduces to
(11)$$\begin{array}{ccc} {\dot{r}}_{z} & = & -1-r_{z}+\left(r_{+}+r_{-}\right)f,\\ {\dot{r}}_{-} & = & -\left(1+i\omega_{12}\right)r_{-}-\left(1+r_{z}\right)f,\\ & = & {\left({\dot{r}}_{+}\right)}^{*}. \end{array}$$In Figure 2, the smooth monotonic decay of the net inversion for no QI (f=0) is replaced in the presence of QI, |f|=1, by “step or plateau decaying” pattern for τ<2, where decreasing decay rate takes place followed by relatively speed up.The transitions between slowing and speeding up occur where the curves for f=1,-1 cross at nearly the same times with the curve of f=0 and are accompanied by maximum peak values of the oscillatory polarization components, which are almost independent of the QI parameter. The effect of the QI on the temporal evolution of the population decay (slowing and speeding) in the present model emulates the Zeno decay (i.e., slowing down) and the anti-Zeno decay (i.e., speeding up) in quantum systems subject to very frequent or not frequent measurement, respectively [26,27]. This is due to the coupling with the net oscillatory polarization in the presence of QI. The real dispersive polarization component, $Re(r_{-}\left(\tau \right))$, in the inset of Figure 2 shows the usual oscillation before setting to its steady state zero-value, independent of the QI parameter. The same is true for the absorptive polarization component, $Im(r_{-}\left(\tau \right)).$**(ii)** 
**The TF case (N≠0,M=0)**
The Bloch equation (Equation (1)) in this case takes the same form as Equation (3), but with the replacement of the unit term (-1) by the broadened term $\left(-N_{1}\right)$, with $N_{1}=1+2N$. The atomic behavior (Figure 3) is similar to the NV case of Figure 2 but with faster decay in the inversion and lesser oscillation in the polarization components due to the broadening effect N≠0 caused by the TF reservoir.For the given value of N≤1, it is noted that the analog of Zeno and anti-Zeno effect occurs less in the case of parallel dipole matrix elements of $p_{l_{1}}$ and $p_{l_{2}}$, i.e., f=1. For larger broadening, N≥1, the purely monotonic decay of the atomic inversion is dominant.**(iii)** 
**The SV case ((N,M≠0)**
The full system of Bloch equation (Equation (1)) has oscillatory terms due to the SV parameter (M≠0). In Figure 4, the net atomic inversion shows faster monotonic decay to its steady oscillations, compared with the NV and TF cases and almost independent of the QI parameter, similar to the one-photon 2-level atom bathed in an SV reservoir outside the rotating wave approximation [28]. On the other hand, both components of the polarizations (inset of Figure 4) show the steady oscillations only due to the presence of the QI parameter, |f|=1.

### 3.2. Wehrl Density and Entropy

Here, the *H*-function and the atomic Wehrl density are used as useful measures for the loss of information in the qubit of two-photon transition model bathed in different radiation reservoirs. The region in which the Wehrl density is independent of the phase space parameters θ and ϕ is called information loss region.

**(i)** 
**The NV reservoir case (M=N=0)**
The Wehrl density as a measure of the amount of information of the qubit resulted in the phase, θ∈[0,4π] and ϕ∈[0,3π], is plotted in Figure 5a–c for different QI parameter f=0 and f=±0.7 and initial symmetric atomic state.The oscillatory behavior along both axes of θ and ϕ, which is an indication that interference is more manifest with negative f=-0.7. The positive peaks resemble those of classical probability distribution, while negative peaks refer to the quantum nature of the system. Note that the negative peaks are relatively deeper for f<0 (as seen by the intense circle areas in Figure 5c).To examine the coherence loss in the present model, we plot the evolution of the Wehrl entropy S(τ) [25,29,30,31] in Figure 6. The monotonic behavior for f=0 becomes non-monotonic due to the QI effect, f≠0.**(ii)** 
**The TF reservoir case (M=0 and N≠0)**
The Wehrl density in this case (Figure 7a–c) shows smaller weights of its positive and negative peaks for f≥0 (Figure 7a,b), compared with the NV case in Figure 5a,b. The case of f>0 shows more flattening of the oscillatory peaks. The negative value of f<0 (Figure 7c) induces deeper negative peaks than in the NV case in Figure 5c.The time evolution of the Wehrl entropy S(τ) (Figure 8) has the same qualitative behavior of the NV case (Figure 6) but with lager amplitudes for f≠0.Note, for larger value of average reservoir photon number N>1, the peaks in both D(θ,ϕ), and S(τ) are flattened due to the broadening effect.**(iii)** 
**The SV reservoir case (M,N≠0)**
The Wehrl density D(θ,ϕ) for f=0 (Figure 9a) resembles that of the NV and TF reservoirs cases (see Figure 5 and Figure 7) but with larger weights of positive and negative peaks. The flattening of these peaks for f>0 (Figure 9b) is relatively more than in the TF reservoir case (Figure 7b).For f<0 (Figure 9c), the negativity of the peaks is more intense and its location in (θ,ϕ) plane are shifted (compared with Figure 5c and Figure 7c).The Wehrl entropy S(τ) in Figure 10 shows almost monotonic behavior for f=0, but irregular oscillatory behavior with larger amplitudes show for non-zero value f≠0.For larger N>1, the Wehrl density becomes more flattened and the qubit loses its coherence, with S(τ) tending faster and monotonically to its stationary value.

### 3.3. Entropy Squeezing

Here, we present the temporal evolutions of the entropy squeezing $E_{y}$ for the initial symmetric state ${|\psi \rangle }^{A}\left(0\right)=\frac{1}{\sqrt{2}}[|↑\rangle +\left|↓\rangle \right]$.

In the NV case (N=M=0), the entropy squeezing $E_{y}\left(\tau \right)$ (Figure 11a) has oscillatory behavior before it reaches its steady state. Squeezing occurs ($E_{y}\left(\tau \right)<0$) with f≠0, with more squeezing with negative f<0 for a short time τ<0.1.

In the TF case of M=0 and N=1 (Figure 11b), squeezing occurs only for negative value of QI parameter f<0. The squeezing is much more in magnitude in the SV case (N=1 and $M=\sqrt{N(N+1)}$) for negative f<0 (see Figure 11c). Thus, in the three cases of radiation reservoirs, entropy squeezing is due to the QI effect.

## 4. Summary

We have examined the model of 2-level atom model with two-photon transition in the presence of different radiation reservoirs, namely, the NV, TF and SV reservoirs. The effects of the quantum interference (QI) parameter on the atomic dynamics, the coherence and information loss in the phase space (utilizing Wehrl entropy and its phase density) and entropy squeezing are analyzed. The main results are: (i) The QI and the radiation reservoir parameters shape the atomic dynamics as monotonic, non-monotonic or oscillatory structure. Specifically, in the NV case, the decay of the atomic inversion shows similarity with the quantum Zeno and anti-Zeno decay phenomena. In the TF and SV cases, radiative line broadening leads to a faster monotonic decay, with persistent steady oscillations in the SV case. The atomic polarization components are essentially insensitive to the QI parameter in all three cases of radiation reservoir states. (ii) Time evolution of the Wehrl’s entropy in the NV and TF cases shows non-monotonic behavior due to the QI process, whilst, in the SV case, it shows irregular oscillatory behavior that flattens with larger line broadening, thus the qubit loses its coherence. On the other hand, in the SV case, Wehrl’s density shows prominent intensive negative peaks for negative value of the QI parameter (f<0) in the atomic coherent state phase parameters (θ,ϕ)-plane: an indication of quasi-probability information. (iii) Depending on the initial atomic state, maximum (Boltzmann’s) entropic squeezing occurs in the SV case with negative value of the QI parameter (f<0).

## Figures and Tables

**Figure 1 entropy-21-00147-f001:**
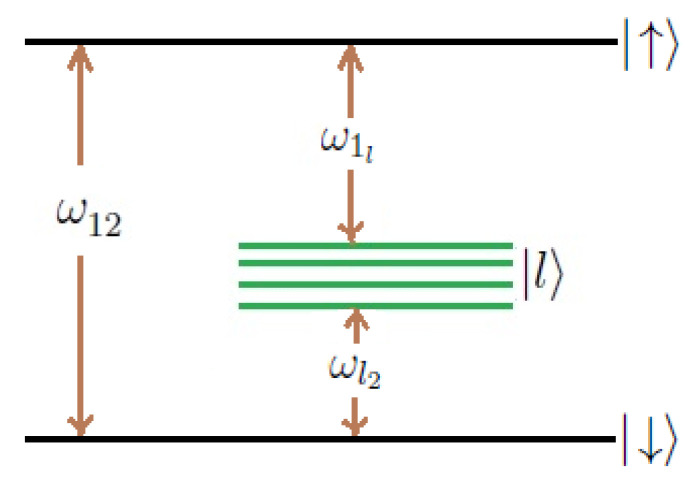
The two-photon transition two-level atomic scheme.

**Figure 2 entropy-21-00147-f002:**
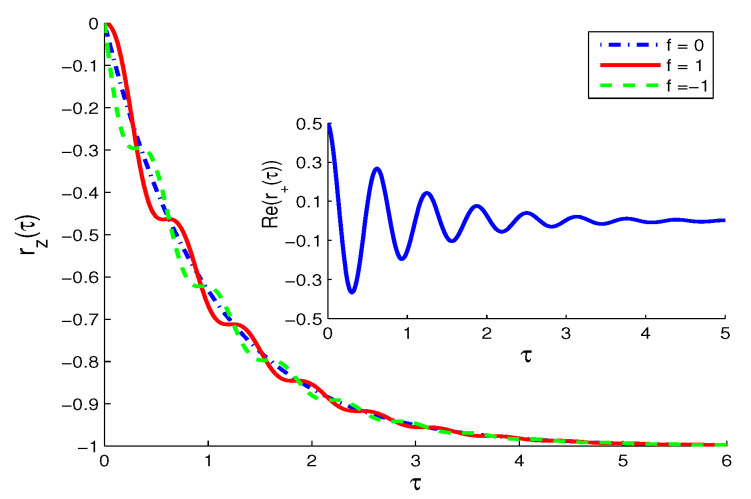
Mean atomic inversion, $r_{z}$ in the normal vacuum (NV) case (N=M=0) for $\omega_{12}=10$ and initial state ${|\psi \rangle }^{A}\left(0\right)=\frac{1}{\sqrt{2}}[|↑\rangle +\left|↓\rangle \right]$ for f=0,±1. Inset shows the atomic polarization component, $Re(r_{+}\left(\tau \right))$, which is almost independent of the QI parameter *f* (the same result is obtained for $Im(r_{+}\left(\tau \right))$.

**Figure 3 entropy-21-00147-f003:**
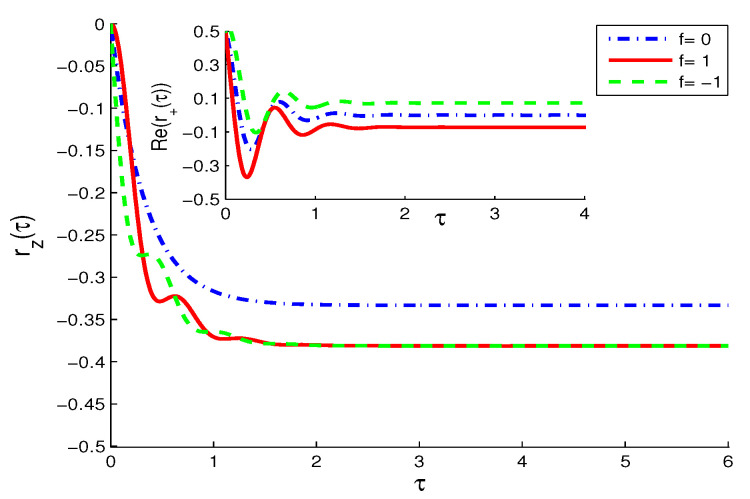
The same as Figure 2 but in the thermal field (TF) case (N=1,M=0).

**Figure 4 entropy-21-00147-f004:**
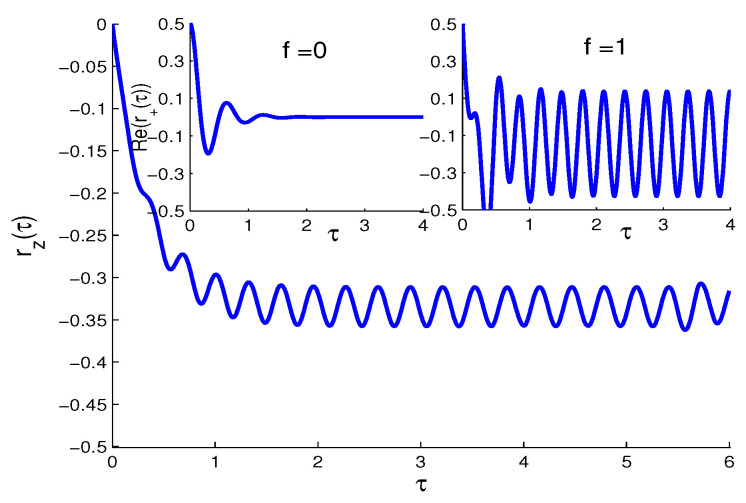
The same as Figure 2 but in the squeezed vacuum (SV) case ($N=1,M=\sqrt{N(N+1)}$). Note that $r_{z}\left(\tau \right)$ is almost independent of the QI parameter *f*. The two insets show $Re(r_{+}\left(\tau \right))$, for f=0,f=1 (same results are obtained for $Im(r_{+}\left(\tau \right))$)

**Figure 5 entropy-21-00147-f005:**
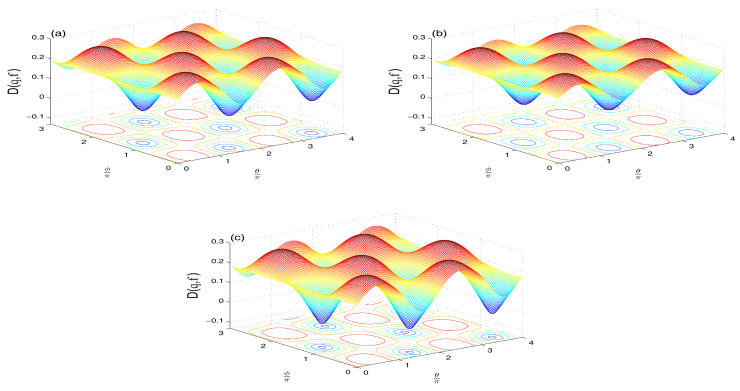
Wehrl density D(θ,ϕ,3π) in the NV case of N=M=0 when ${|\psi \rangle }^{A}\left(0\right)=\frac{1}{\sqrt{2}}[|↑\rangle +\left|↓\rangle \right]$ with $\omega_{12}=10$ and: f=0 (**a**); f=0.7 (**b**); and f=−0.7 (**c**).

**Figure 6 entropy-21-00147-f006:**
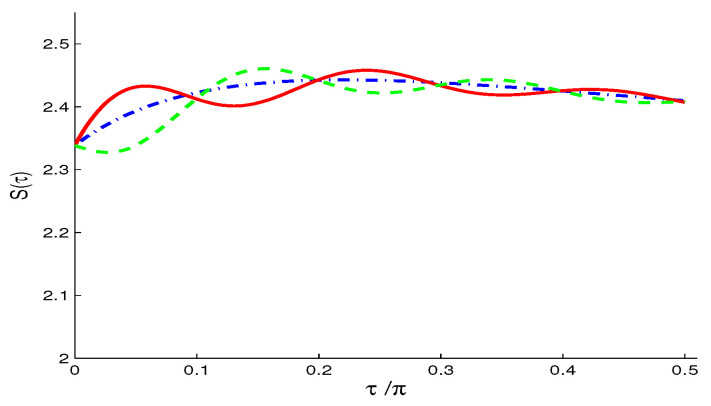
Wehrl entropy S(τ), in the NV case of M=N=0 and different values of f=0 (dashed-dotted curve), f=0.7 (solid curve) and f=−0.7 (dashed curve) when ${|\psi \rangle }^{A}\left(0\right)=\frac{1}{\sqrt{2}}[|↑\rangle +\left|↓\rangle \right]$, with $\omega_{12}=10$.

**Figure 7 entropy-21-00147-f007:**
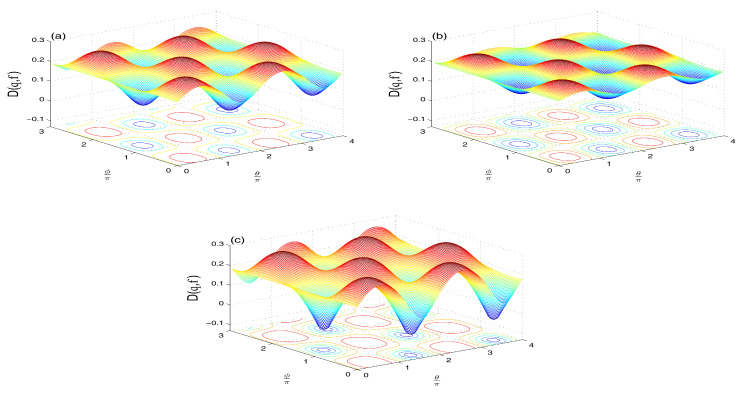
Wehrl density D(θ,ϕ,3π) in the TF case of M=0 and N=1 when ${|\psi \rangle }^{A}\left(0\right)=\frac{1}{\sqrt{2}}[|↑\rangle +\left|↓\rangle \right]$ with $\omega_{12}=10$ and: f=0 (**a**); f=0.7 (**b**); and f=−0.7 (**c**).

**Figure 8 entropy-21-00147-f008:**
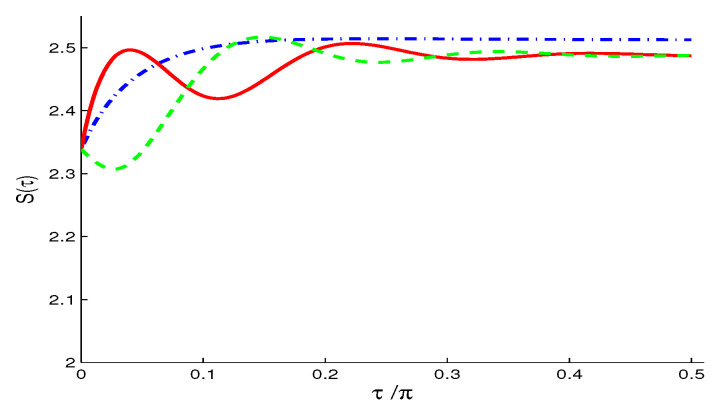
Wehrl entropy S(τ), in the TF case of M=0 and N=1 and different values of f=0 (dashed-dotted curve), f=0.7 (solid curve) and f=−0.7 (dashed curve) when ${|\psi \rangle }^{A}\left(0\right)=\frac{1}{\sqrt{2}}[|↑\rangle +\left|↓\rangle \right]$, with $\omega_{12}=10$.

**Figure 9 entropy-21-00147-f009:**
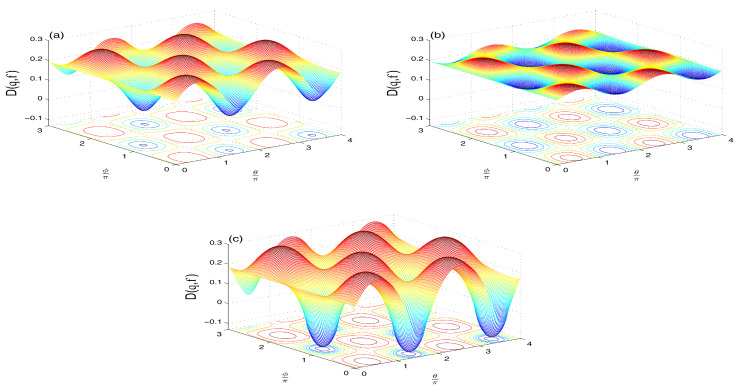
Wehrl density D(θ,ϕ,3π) in the SV case of N=1 and $M=\sqrt{N(N+1)}$ when ${|\psi \rangle }^{A}\left(0\right)=\frac{1}{\sqrt{2}}[|↑\rangle +\left|↓\rangle \right]$ with $\omega_{12}=10$ and: f=0 (**a**); f=0.7 (**b**); and f=−0.7 in (**c**).

**Figure 10 entropy-21-00147-f010:**
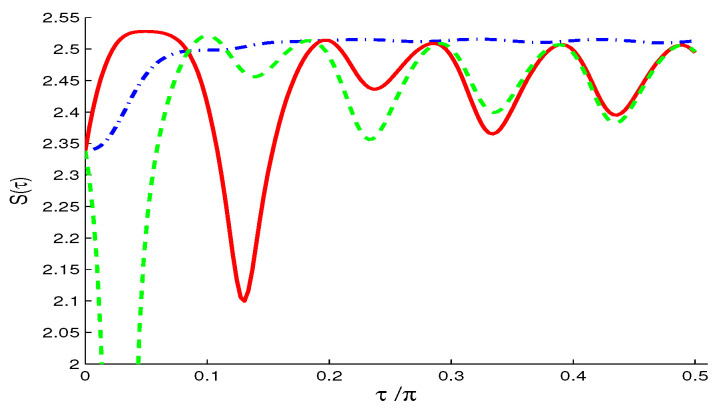
Wehrl entropy S(τ), in the SV case of N=1 and $M=\sqrt{N(N+1)}$ and different values of f=0 (dashed-dotted curve), f=0.7 (solid curve) and f=−0.7 (dashed curve) when ${|\psi \rangle }^{A}\left(0\right)=\frac{1}{\sqrt{2}}[|↑\rangle +\left|↓\rangle \right]$, with $\omega_{12}=10$.

**Figure 11 entropy-21-00147-f011:**
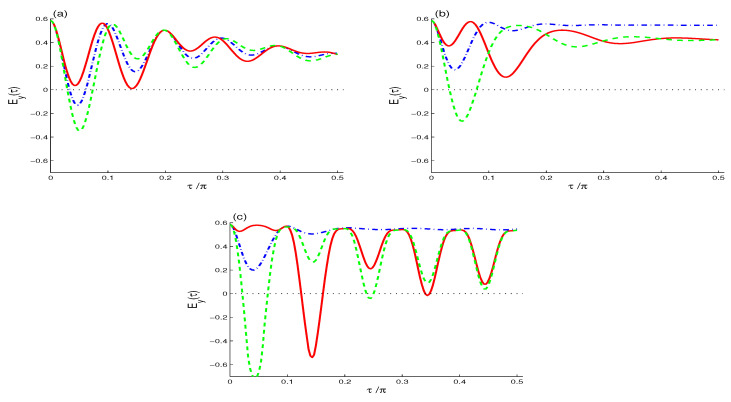
The entropy squeezing $E_{y}\left(\tau \right)$ when ${|\psi \rangle }^{A}\left(0\right)=\frac{1}{\sqrt{2}}[|↑\rangle +\left|↓\rangle \right]$. For different values of f=0 (dashed-dotted curve), f=0.7 (solid curve) and f=−0.7 (dashed curve) when $\omega_{12}=10$: the NV case (**a**); the TF case (**b**); and the SV case (**c**).

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
