# Peer review of "Quantum Interference Effects on Information Phase Space and Entropy Squeezing"

_entropy, 2019, doi:10.3390/e21020147_

Round 1

Reviewer 1 Report

The Authors discuss an atomic model involving two-photon transitions. Applying the Bloch equations, they have numerically found the time-evolution of mean atomic inversion, Wherl entropy, its density, and the entropy squeezing. They considered the cases when the system interacts with the vacuum, thermal and squeezed vacuum environments.

The article is written in the right way – all calculations and results are presented clearly. In my opinion, the manuscript contains potentially valid results, but the Authors should convince the Readers about that fact. In particular, the Conclusion section should be extended considerably. The Authors should include there the discussion of their results, especially in the context of the influence of various types of the external bath on the appearance of quantum correlations appearing in the system. Moreover, some of the topics discussed in the article are those related to the Wehrl entropy. The latter can be applied as a tool in the research concerning quantum correlations. That fact was already discussed in the papers J. Phys. A 34:4951 (2001) and J. Phys. A 34:3887 (2001). Adding at least a short paragraph (in subsection 4.1) which would mention those facts, will be highly desirable from the educational point of view.

Therefore I recommend accepting the manuscript for the publication after amendments along the points listed above.

Author Response

The referees are thanked for their detailed valuable comments. In the modified version, we have responded to the referees comments. Specifically:

(i) We have extended the conclusion to mention the  main finding of our work.

(ii) The reference to Wehrl entropy is now in the new Sec.2 (B) (the old Sec.4.1) with new added [22-24].

Reviewer 2 Report

The authors present a parametric analysis of coherence and information loss in phase space for a two-photon two-level atom system coupled to a squeezed vacuum. The system is described by a model based on Bloch equations, derived by the one of the authors in a previous publication (namely Ref. 4). From the solution to these equations, the authors determine the effect quantum interference (quantified by what they call a quantum interference normalized parameter) in terms of different configurations of the squeezed vacuum. The analysis is then followed by the definition (within the context) and computation of quantities such as the Wehrl entropy and density, or the Boltzmann entropy.

In the present form, the manuscript has only marginal interest, since it is rather vague. However, it could be remarkably improved and, therefore, be suitable for publication by introducing a series of changes:

1) The introduction, as well as an important part of the work, essentially makes reference to previous work by the same authors. Although I am not against this practice, which I consider a nice way to follow the evolution of a given research line or a problem by a researcher or a collaboration, introductions providing a wider perspective on the issue are also highly recommended. In the present case, this is an important flaw that the authors should carefully reconsider.

2) Even if the model has already been described in another work, at least the basic features and/or hypothesis upon which such models are built on should be mentioned, something that does not happen here. Specifically, authors start the work directly with the set of Bloch equations (1) without any previous explanation of the model. Even if the derivations involved are relatively simple to reproduce, I think it would help the readability and dissemination of the work.

3) The organization of the work is also a bit, say, funny, with some results in between two theoretical sections. If you accept a suggestion in this regard, Secs. 2 and 4 should be merged together, as a single theoretical section containing all the theory used and discussed in the work, while Sec. 3 should be included into Sec. 5, thus gathering all computations.

4) The computational results need a deeper physical discussion and not only a few descriptive lines of what the reader can already see in the figures. For example, something I have observed is that, in Fig. 1, the crossing between the green and red curves at the point where the latter reaches a plateau coincides (or apparently it seems to coincide) with the blue curve. Is this a general trend for any NV case with N = M = 0? This is interesting, because previous to that crossing the red curve seems to undergo Zeno delay followed by anti-Zeno decay. This anti-Zeno behavior arises after the previous crossing with the green curve, which behaves just in the opposite way. And this overall behavior continues after each crossing of both curves with the blue one. On the other hand, there is a direct relationship between these oscillations and those observed in the inset: each time the three colored curves meet, there is a maximum in the inset curve, while its minima seem to appear whenever the green and red lines cross each other. In the other two cases, although it is more complicated, the same (or something similar) can be applied. So, in summary, I miss a deeper discussion of the physics described by the figures, not only in Sec. 3, but that can also be extended to the computations of Sec. 5.

5) The final section looks more a sort of general summary of the work than a true concluding section, providing the reader with a valuable take-home message.

The grammar should also be checked (as well as the English). Some examples:

a) The sentence in lines 32 and 33 makes no sense. It has to be checked, because it probably arises from a previous version.

b) There is an opening parenthesis in line 38 that should not be there (unless there is a closing missing parenthesis somewhere else).

c) The period in line 104 should be a comma.

Author Response

The referees are thanked for their detailed valuable comments. In the modified version, we have responded to the referees comments. Specifically:

(i) The 1st paragraph of introduction, now refers to the QI process and its implications, with new added [3-11], experimentally and theoretically.

(ii) The main physical assumptions involved in the model Bloch eqs are added in the last paragraph of the new Sec.2(A).

(iii) We have merged the old Sec.2 and 4 to become the new Sec.2, and the old Sec.3 and 5 to become the new Sec.3.

(iv) The physical discussion regarding the atomic inversion decay and  its analogy with the quantum Zeno and anti-Zeno effect is inserted in the new Sec.III (A), in the NV case (we thank the referee of this valuable remark in our model), with new added [26,27].

(v) The summary is extended to include our main results in some detail.

Reviewer 3 Report

It might be worthwhile to mention that the Wehrl entropy is not invariant under all unitarian transformations in contrast to the von Neumann entropy.Moreover, the Wehrl entropy is strictly lower bounded by the von Neuman theory.The Wehrl entropy is positive ( as is the von Neuman entropy) and it should be rememberd that the minimum value for the Werhl entropy is 1.

Author Response

The referees are thanked for their detailed valuable comments. In the modified version, we have responded to the referees comments. Specifically:

(i) We have extended the summary section (as well the introduction) accordingly.

(ii) The refer to Wehrl entropy and some of its mathematical property is added in the new Sec.2 (B) with  new added [22-24].

Round 2

Reviewer 2 Report

I would like to thank the remarkable effort made by the authors in improving their manuscript. I feel satisified with this revised version and recommend its publication in Entropy in its current form.